# Enhancing Medical Imaging Segmentation with GB-SAM: A Novel Approach to Tissue Segmentation Using Granular Box Prompts

**DOI:** 10.3390/cancers16132391

**Published:** 2024-06-28

**Authors:** Ismael Villanueva-Miranda, Ruichen Rong, Peiran Quan, Zhuoyu Wen, Xiaowei Zhan, Donghan M. Yang, Zhikai Chi, Yang Xie, Guanghua Xiao

**Affiliations:** 1Quantitative Biomedical Research Center, Peter O’Donnell Jr. School of Public Health, University of Texas Southwestern Medical Center, Dallas, TX 75390, USA; ismael.villanueva-miranda@utsouthwestern.edu (I.V.-M.); ruichen.rong@utsouthwestern.edu (R.R.); peiran.quan@utsouthwestern.edu (P.Q.); zhuoyu.wen@utsouthwestern.edu (Z.W.); xiaowei.zhan@utsouthwestern.edu (X.Z.); donghan.yang@utsouthwestern.edu (D.M.Y.); yang.xie@utsouthwestern.edu (Y.X.); 2Department of Pathology, University of Texas Southwestern Medical Center, Dallas, TX 75390, USA; zhikai.chi@utsouthwestern.edu; 3Simmons Comprehensive Cancer Center, University of Texas Southwestern Medical Center, Dallas, TX 75390, USA; 4Department of Bioinformatics, University of Texas Southwestern Medical Center, Dallas, TX 75390, USA

**Keywords:** digital pathology, pathology image, segmentation, foundation models, histopathology

## Abstract

**Simple Summary:**

The complexity and diversity of tissue structures in histopathology present significant challenges for accurately segmenting images, which is crucial for cancer diagnosis. Our research introduces the Granular Box Prompt Segment Anything Model (GB-SAM), developed to improve segmentation accuracy with less dependence on extensive annotated datasets. GB-SAM aims to precisely segment glandular structures using granular box prompts, essential for accurate cancer detection. In our experiments, GB-SAM consistently outperformed traditional models like U-Net, achieving a Dice coefficient of 0.885 on the CRAG dataset with only 25% of the training data, compared to U-Net’s 0.857. This advancement can significantly simplify diagnostic workflows in digital pathology, offering a robust tool for researchers and clinicians and potentially transforming clinical practices to improve patient outcomes.

**Abstract:**

Recent advances in foundation models have revolutionized model development in digital pathology, reducing dependence on extensive manual annotations required by traditional methods. The ability of foundation models to generalize well with few-shot learning addresses critical barriers in adapting models to diverse medical imaging tasks. This work presents the Granular Box Prompt Segment Anything Model (GB-SAM), an improved version of the Segment Anything Model (SAM) fine-tuned using granular box prompts with limited training data. The GB-SAM aims to reduce the dependency on expert pathologist annotators by enhancing the efficiency of the automated annotation process. Granular box prompts are small box regions derived from ground truth masks, conceived to replace the conventional approach of using a single large box covering the entire H&E-stained image patch. This method allows a localized and detailed analysis of gland morphology, enhancing the segmentation accuracy of individual glands and reducing the ambiguity that larger boxes might introduce in morphologically complex regions. We compared the performance of our GB-SAM model against U-Net trained on different sizes of the CRAG dataset. We evaluated the models across histopathological datasets, including CRAG, GlaS, and Camelyon16. GB-SAM consistently outperformed U-Net, with reduced training data, showing less segmentation performance degradation. Specifically, on the CRAG dataset, GB-SAM achieved a Dice coefficient of 0.885 compared to U-Net’s 0.857 when trained on 25% of the data. Additionally, GB-SAM demonstrated segmentation stability on the CRAG testing dataset and superior generalization across unseen datasets, including challenging lymph node segmentation in Camelyon16, which achieved a Dice coefficient of 0.740 versus U-Net’s 0.491. Furthermore, compared to SAM-Path and Med-SAM, GB-SAM showed competitive performance. GB-SAM achieved a Dice score of 0.900 on the CRAG dataset, while SAM-Path achieved 0.884. On the GlaS dataset, Med-SAM reported a Dice score of 0.956, whereas GB-SAM achieved 0.885 with significantly less training data. These results highlight GB-SAM’s advanced segmentation capabilities and reduced dependency on large datasets, indicating its potential for practical deployment in digital pathology, particularly in settings with limited annotated datasets.

## 1. Introduction

The latest advancements in digital pathology present significant practical benefits compared to traditional manual diagnosis [1,2]. The exponential growth of medical imaging technologies has resulted in an accumulation of high-resolution histological images, necessitating automated annotation and diagnosis processes [3]. In this context, artificial intelligence (AI) algorithms have emerged as promising tools in digital pathology, holding immense potential for streamlining diagnostic workflows [4,5].

However, the diversity of biological tissue structures complicates the automated analysis of histopathology slides [6,7]. This challenge is particularly pronounced in the segmentation of tissue and substructures, such as glands and nodules, which are pivotal in cancer diagnosis across various tissue types and cancer subtypes [8,9]. Disruptions in glandular structures are often indicative of malignant cases [10], presenting irregular shapes in contrast to the circular structures commonly observed in benign cases. This distinction is crucial in diagnosing colorectal cancer, where the gland anatomy plays a vital role [11,12]. Therefore, the precision and accuracy of tissue segmentation processes are critical for advancing toward an AI-aided cancer detection pipeline.

The AI research community is currently experiencing a significant transformation, driven by the development of large-scale models like DALL-E [13], GPT-4 [14], and SAM [15]. These models provide frameworks for solving a wide range of problems. SAM, in particular, emerges as a notable segmentation model capable of generalizing to new objects and images without further training. This adaptability results from SAM’s training on millions of images and masks, refined through iterative feedback and model improvements. However, in order to apply SAM in the medial field, it is necessary to fine-tune it for a given downstream task.

The contributions of our work are the following:Introducing a new strategy for fine-tuning the SAM model using granular box prompts derived from ground truth masks, enhancing gland morphology segmentation accuracy.Demonstrating through experiments on CRAG, GlaS, and Camelyon16 datasets, our training strategy improves SAM’s segmentation performance.Showcasing SAM’s superior performance and adaptability in digital pathology is particularly beneficial for cases with limited data availability.Highlighting SAM’s consistent performance and exceptional ability to generalize to new and complex data types, such as lymph node segmentation.

This paper is organized as follows: In Section 2, we discuss related work in the field of medical image segmentation using SAM. Section 3 describes the datasets and the methodology used, including the training procedures for GB-SAM. Section 4 presents our experimental results and provides a comparative analysis with other models. Finally, Section 5 concludes the paper, summarizing our findings and discussing future research directions.

## 2. Related Work

The application of the Segment Anything Model (SAM) in the pathology field remains relatively unexplored. SAM-Path improves SAM’s semantic segmentation in digital pathology by introducing trainable class prompts. Experiments on the BCSS and CRAG datasets show significant improvements over the standard SAM model, highlighting its enhanced performance in pathology applications [16].

Another study evaluated SAM’s zero-shot segmentation on whole slide imaging (WSI) tasks, such as tumor, nontumor tissue, and cell nuclei segmentation. While SAM obtained good results in segmenting large connected objects, it struggled with dense instance object segmentation [17].

WSI-SAM focuses on precise object segmentation for histopathology images using multiresolution patches. This variant maintains SAM’s zero-shot adaptability and introduces a dual mask decoder to integrate features at multiple scales, demonstrating superior performance on tasks like ductal carcinoma in situ (DCIS) and breast cancer metastasis segmentation [18].

MedSAM, a foundation model for universal medical image segmentation, covers various imaging modalities and cancer types. It outperforms modality-wise specialist models in internal and external validation tasks, indicating its potential to revolutionize diagnostic tools and treatment plans [19].

Another paper explores SAM’s application in medical imaging, particularly radiology and pathology. Through fine-tuning, SAM significantly improves segmentation accuracy and reliability, offering insights into its utility in healthcare [20].

Lastly, all-in-SAM, an SAM pipeline for the AI development workflow, has shown that leveraging pixel-level annotations from weak prompts can enhance the SAM segmentation model. This method surpasses state-of-the-art methods in nuclei segmentation and achieves competitive performance with minimal annotations [21].

We summarize these works in Table 1.

## 3. Material and Methods

### 3.1. Datasets

#### 3.1.1. CRAG for Training and Validation

The CRAG dataset [22] is our model’s internal training and validation foundation. It consists of 213 H&E-stained histopathology images, specifically on colorectal adenocarcinoma tissues. The dataset is divided into 173 images dedicated to training the model and 40 images for testing its accuracy. Each image is annotated at the pixel level to delineate the glandular structures precisely. This detailed annotation allows a robust evaluation of the model’s gland segmentation capabilities. At the same time, the image resolution varies within the dataset, typically around 20× magnification. Samples from the CRAG dataset can be seen in Figure 1a.

#### 3.1.2. External Testing Datasets

We selected the GlaS and Camelyon16 datasets to evaluate GB-SAM trained on CRAG.

GlaS Dataset: The GlaS dataset [23], an external testing benchmark, enhances the generalizability of our model’s performance beyond the training data. It offers 165 H&E-stained images derived from colon tissue sections, providing a broader range of histological features compared to CRAG. The dataset is divided into 85 training images and 80 testing images. Like CRAG, images are meticulously annotated at the pixel level to identify glandular structures. The GlaS dataset is a popular choice for evaluating gland segmentation algorithms due to the exceptional quality of its annotations and the inclusion of images with diverse gland morphologies, reflecting the variations observed across different histological grades. This variety ensures that the model encounters broader challenges during testing. Samples from the GlaS dataset can be seen in Figure 1b.

Camelyon16 Dataset: The Camelyon16 dataset [24] is important in assessing our model’s ability to detect lymph node metastasis in breast cancer patients. It includes a collection of 400 whole-slide images (WSIs) obtained from sentinel lymph nodes, the first lymph nodes to receive drainage from a tumor site. The dataset is further split into 270 training slides and 130 testing slides. Each WSI is annotated to mark the exact regions containing metastatic cancer cells. Samples from the Camelyon dataset can be seen in Figure 1c.

#### 3.1.3. Datasets Representation

Let $D=\{d_{1},d_{2},\ldots ,d_{n}\}$ represent a dataset, where each $d_{i}$ is a data item. Each $d_{i}$ includes an image $I_{i}$ and a set of objects $O_{i}=\left\{o_{i1},o_{i2},\ldots ,o_{ij}\right\}$ associated with the image. Each object $o_{ij}$ is accompanied by annotations, including bounding boxes (*B*) and ground truth masks (*M*).

### 3.2. Granular Box Prompts SAM

#### 3.2.1. Segment Anything Model

In our approach to fine-tuning GB-SAM, we chose the SAM-ViT-Huge version as our starting point. This version delivers high-quality segmentation but incurs higher running time and memory usage due to its large ViT architecture [25]. However, we aimed to customize GB-SAM for our specific segmentation tasks, which required high precision and segmentation quality.

To meet our fine-tuning goals, we disabled gradient computation for both the vision encoder and prompt encoder components (See Figure 2). This step is crucial to keep the weights of these components unchanged during the fine-tuning process, thus maintaining the integrity of SAM’s pre-trained capabilities.

Subsequently, the fine-tuning process unfolds through a sequence of steps, including loss function minimization, parameter optimization, adaptive learning rate adjustment, and performance evaluation.

#### 3.2.2. Object Selection and Bounding Box Prompt

We employ a strategy that uses granular small box prompts derived from ground truth masks instead of a single large box prompt covering the entire H&E-stained image patch (see Figure 2). This approach allows a more localized and detailed understanding of gland morphology, aligning closely with biological tissues’ inherent heterogeneity and irregularity.

This technique also supplies GB-SAM with highly granular data, enhancing the segmentation accuracy of individual glands by mitigating the ambiguity a larger box might introduce, especially in morphologically complex regions.

Moreover, smaller boxes enable the model to concentrate on specific gland features, supporting a more robust learning process and potentially boosting the model’s ability to generalize across diverse histopathological patterns in H&E-stained images.

To facilitate this, we define f(j) as the recursive function for retrieving an object *O* from an image mask *m* at the random index *j*, where *j* ranges from 0 to n−1, and *n* represents the total number of objects in the image mask.
$$f\left(j\right)=\left\{\begin{array}{cc} \mathrm{getObject}(j) & \mathrm{if}\;\mathrm{getObject}(j)\neq \mathrm{None}\\ f(j+1) & \mathrm{if}\;\mathrm{getObject}(j)=\mathrm{None}\\ & \;\mathrm{and}\;j+1<n\\ f(j-1) & \mathrm{if}\;\mathrm{getObject}(j)=\mathrm{None}\\ & \;\mathrm{and}\;j+1\geq n \end{array}\right.$$

The retrieval process involves using the function getObject(j) to directly access objects in the image mask and perform augmentations operations. If no valid objects are found, the function selects a new random object from the valid ones and adjusts its mask.

Finally, for each selected object $o_{ij}$, a bounding box prompt *B* is generated based on $o_{ij}$’s bounding box, with additional random adjustments, thus generating processing inputs for GB-SAM.

#### 3.2.3. Training Procedures

#### Image Preprocessing and Augmentation

We applied a comprehensive image augmentation strategy to enhance the robustness of our GB-SAM model against variations in H&E-stained images. This approach, formulated through probabilistically selected transformations, aims to simulate a diverse array of histopathological image conditions. Our augmentation process can be represented as follows:

Let $T=\{T_{1},T_{2},\ldots ,T_{k}\}$ be a set of transformation functions, where each $T_{i}$ corresponds to a specific image augmentation technique. Each transformation $T_{i}$ is selected with a probability $p_{i}$ = 0.5. Then, the probabilistic selection is denoted by $T_{selected}=Select\left(T_{i},p_{i}\right)$.

The augmentation transformations include noise addition and blur, spatial transformations (flipping, shifting, scaling, and rotating), and morphological distortions (elastic, grid, and optical). Additionally, adjustments to color channels $\left(T_{rgb}\right)$ and brightness and contrast levels $\left(T_{bc}\right)$ are consistently applied to each object.

Hence, the comprehensive augmentation function encompassing all specific transformations applied to an object *O* is represented as
$$A\left(O\right)=\left(T_{\mathrm{rgb}}⊕T_{\mathrm{bc}}⊕\overset{k}{\underset{i=1}{⨁}}T_{\mathrm{selected}}\right)\left(O\right)$$

#### Model Optimizations

Loss Function Minimization

At the core of our training methodology is the objective to minimize a composite loss function, L(θ), which combines the Dice loss [26] and cross-entropy loss [27], defined as follows:$$L\left(\theta \right)=\left(1-\mathrm{Dice}\left(M,M^{*}\right)\right)+\lambda \cdot \mathrm{CE}\left(M,M^{*}\right)$$
where:$\mathrm{Dice}\left(M,M^{*}\right)=\frac{2\cdot |M\cap M^{*}|}{\left|M|+\right|M^{*}|}$ measures the overlap between the predicted and ground truth masks.$\mathrm{CE}\left(M,M^{*}\right)=-\sum_{i}M_{i}^{*}\log\left(M_{i}\right)+\left(1-M_{i}^{*}\right)\log\left(1-M_{i}\right)$ represents the cross-entropy loss, penalizing the pixel-wise differences between the predicted mask *M* and the ground truth $M^{*}$.λ is a balancing coefficient.

The balancing coefficient, λ, is essential for modulating the influence of each loss component, ensuring a balanced optimization focus that addresses global shape alignment and local pixel accuracy. Furthermore, we have configured the loss function to use sigmoid activation and squared predictions, averaged across the batch, to facilitate a refined approach to minimizing prediction errors.

Learning Rate Adjustment

We incorporated the ReduceLROnPlateau strategic learning rate adjustment mechanism [28] into our training process, which is defined as
$$\alpha_{t+1}=\left\{\begin{array}{cc} \alpha_{t}\cdot \eta & \mathrm{if}\;\Delta L_{\mathrm{val}}\leq \varepsilon \;\mathrm{for}\;\mathrm{patience}\;\mathrm{epochs}\\ \alpha_{t} & \mathrm{otherwise} \end{array}\right.$$
where

η<1 is the reduction factor.$\Delta L_{\mathrm{val}}$ measures the change in validation loss.ε is a threshold for determining if the change in loss is significant.

In general, this mechanism adjusts the learning rate α, depending on the performance of GB-SAM on the validation set. We set the reduction factor η at 0.001, which reduces the learning rate after a nonimprovement period of 10 epochs and ϵ was set to 0.0001.

Early Stopping

Incorporating an early stopping mechanism addresses the critical concern of overfitting by monitoring the validation loss $L_{\mathrm{val}}$. Training is halted if GB-SAM stops to show significant improvement over a set number of epochs.
$$\mathrm{if}\;\Delta L_{\mathrm{val}}\leq \varepsilon \;\mathrm{for}\;\mathrm{patience}\;\mathrm{epochs},\;\mathrm{stop}\;\mathrm{training}.$$

This decision is grounded in the observation that continued training beyond this point does not yield substantial gains in validation performance and may, in fact, weaken the GB-SAM generalization capability due to overfitting to the training data.

### 3.3. Comparison and Evaluation

#### 3.3.1. Compared Methods: U-Net, Path-SAM, Med-SAM

#### U-Net Model

To compare the performance of GB-SAM against a well-established segmentation model, we developed and trained a U-Net model. We utilized the U-Net model with a ResNet34 backbone [29], pretrained on the ImageNet dataset [30]. During preprocessing, images were resized to the dimensions of 1024 × 1024 and subjected to augmentation techniques, including noise addition, flipping, rotation, and distortion. Subsequently, each image underwent self-normalization using Z-score normalization. For optimization, we employed the AdamW optimizer with a consistent learning rate of 0.0001 and weight decay of 0.005 for the CRAG dataset. The loss function used was cross-entropy loss. Model training consisted of 60 epochs, with additional parameters including a batch size of 4 and 4 num_workers.

#### SAM-Path

We use the reported results of SAM-Path—a model that employs trainable class prompts alongside a specialized pathology encoder—to compare the performance of our GB-SAM version. SAM-Path enhances the original SAM model’s proficiency in executing semantic segmentation tasks without requiring manual input prompts [16].

#### MedSAM: Segment Anything in Medical Images

In our work, we leveraged the scores obtained and reported from MedSAM for gland segmentation—a foundational model designed for universal medical image segmentation—as a benchmark to evaluate the performance of our GB-SAM model. MedSAM aims for universal segmentation across various tasks and utilizes a large dataset that covers multiple imaging modalities and cancer types [19].

#### Evaluation Metrics

To evaluate the performance of GB-SAM on gland-like segmentation in H&E images, we employ three key metrics: intersection over union (IoU), Dice similarity coefficient (DSC), and mean average precision (mAP). These metrics collectively assess the accuracy of gland boundary delineation and the reliability of gland segmentation.

#### 3.3.2. Intersection over Union (IoU)

The IoU metric quantifies the overlap between the predicted and ground-truth segmentation masks. This metric directly evaluates segmentation accuracy, highlighting both correct and incorrect predictions. However, it may not fully detail the performance in segmenting very small or intricately shaped glands, where precise shape correspondence is crucial.
$$\mathrm{IoU}=\frac{|M\cap M^{*}|}{|M\cup M^{*}|}$$
where *M* is the ground truth mask and $M^{*}$ is the predicted mask.

#### 3.3.3. Dice Similarity Coefficient (DSC)

DSC assesses the similarity between predicted and ground truth masks, focusing on the accuracy of size and shape. This emphasis makes DSC especially valuable for medical segmentation tasks, where precise delineation of structures is critical. Due to its sensitivity to size matching, DSC is often preferred for segmenting small objects.
$$\mathrm{DSC}=\frac{2\;\times \;|M^{*}\cap M|}{|M^{*}\left|\;+\;|M\right|}$$
where *M* is the ground truth mask and $M^{*}$ is the predicted mask.

#### 3.3.4. Mean Average Precision (mAP)

mAP measures the model’s capability to detect and accurately segment distinct objects or classes, calculating the mean of average precision (AP) scores across classes or instances. mAP provides an overview of a model’s detection and segmentation precision, including its success in accurately identifying and outlining objects. This metric is important for multiclass segmentation tasks, as it evaluates detection accuracy and precision across varying confidence levels, unlike IoU and DSC, which focus on geometric overlap and similarity.
$$\mathrm{mAP}=\frac{1}{N}\sum_{i=1}^{N}{\mathrm{AP}}_{i}$$
where *N* is the number of classes, and $AP_{i}$ is the average precision for class *i*.

## 4. Results and Discussion

In this section, we present the experiments conducted to evaluate the segmentation performance of our GB-SAM model and compare its results with those of the U-Net model. Both models were initially fine-tuned using the CRAG dataset, followed by testing on the CRAG testing dataset, GlaS, and Camelyon16 datasets. Additionally, we explore the impact of reduced training dataset sizes. Further details on our experiments and results are provided in the following sections.

### 4.1. Impact of Dataset Size on Tuning GB-SAM and U-Net Models with CRAG

We selected 100%, 50%, and 25% of the CRAG training dataset for our training. We then trained our GB-SAM and U-Net on these subsets and compared the models to the CRAG testing (validation) dataset.

#### 4.1.1. Comparative Results

Analyzing the results obtained from both models, it is noticeable that distinct differences exist in their sensitivity to reductions in training dataset size, as shown in Table 2. The U-Net model exhibits a pronounced decrease in all evaluated performance metrics (Dice, IoU, mAP) as the dataset size decreases, with Dice scores dropping from 0.937 at full dataset size to 0.857 at 25%, IoU scores from 0.883 to 0.758, and mAP scores from 0.904 to 0.765. This trend underscores U-Net’s significant dependency on larger amounts of training data for optimal performance, as highlighted by the standard deviations in performance metrics (Dice: 0.041, IoU: 0.064, mAP: 0.075), indicating variability with changes in dataset size.

In contrast, the GB-SAM model demonstrates a less consistent pattern of performance degradation across the same metrics. Notably, specific metrics (Dice, IoU) even show improvement when the dataset size is reduced to 25%, with Dice scores slightly decreasing from 0.900 at the entire dataset to 0.885 at 25% of the dataset and IoU scores decreasing from 0.813 to 0.793. The average mAP score experiences a modest decline from 0.814 at full dataset size to 0.788 at 25%, demonstrating a less pronounced drop than U-Net. These results suggest that GB-SAM has superior capabilities in generalizing from limited data or experiencing less performance loss due to overfitting on smaller datasets. The low standard deviations for GB-SAM (Dice: 0.012, IoU: 0.016, mAP: 0.019) underscore its consistent performance across varying dataset sizes, in contrast to U-Net’s performance.

Notably, the performance degradation from 100% to 25% dataset sizes was more acute for the U-Net model, especially in mAP scores, indicating a sharper decline in model precision (↓0.139) compared to the GB-SAM model (↓0.026).

It is important to note that while U-Net outperformed GB-SAM when trained on 100% and 50% of the CRAG training dataset, GB-SAM’s results are characterized by more excellent stability, outperforming U-Net when using only 25% of the training data (see Table 2).

#### 4.1.2. Segmentation Performance

Upon analyzing the performance metrics of the GB-SAM and U-Net models across varying training dataset sizes, we identified H&E patch images that exhibited the lowest scores for each training size (Table 3), offering a detailed view of each model’s segmentation capability.

A recurring observation in Table 3 is the challenge presented by the image test_23 to the GB-SAM model, which faces difficulties across all metrics (Dice, IoU, mAP) and dataset sizes. As demonstrated in Figure 3d, GB-SAM’s segmentation is noticeably noisy, showing a tendency toward higher false positives and reduced accuracy, as evidenced by its over-segmentation and unnecessary noise alongside actual features.

In contrast, U-Net’s segmentation more closely matches the ground truth but lacks finer details, missing smaller features and resulting in smoother edges. This indicates U-Net’s better performance in capturing the overall structure in this image, though it struggles to capture detailed aspects (Figure 3c).

In our analysis, we computed the differences between the ground truth and the predicted masks. The segmentation results for GB-SAM and U-Net on image test_23, as shown in Figure 4 and Figure 5, reveal several key observations. The color-coded segmentation, with red indicating underpredictions (where the predicted mask has a pixel as 0 and the true mask has it as 1) and green highlighting overpredictions (where the predicted mask has a pixel as 1 and the true mask has it as 0), allows for a visual comparison of model performance across different training dataset sizes. GB-SAM consistently balances over- and underpredictions, regardless of the training dataset size. This consistency shows that GB-SAM maintains stable segmentation performance, which indicates its robustness and ability to generalize from the available data.

On the other hand, U-Net tends to increase overpredictions as the size of the training dataset decreases. This pattern points to a potential overfitting issue with U-Net when trained on smaller datasets, where the model might compensate for the lack of data by overestimating the presence of features. It also reflects U-Net’s sensitivity to training dataset size, suggesting that its performance and accuracy in segmenting specific features become less reliable with reduced data availability.

For added context and based on Table 3, U-Net struggles with several images (test_39, test_15, test_18) across various training sizes. Notably, U-Net tends to produce gland structures hallucinations, as illustrated in Figure 6, and incorrectly segments scanner artifacts as glandular tissue, as demonstrated in Figure 7.

Our findings show the importance of dataset size in training segmentation models and reveal distinct characteristics of GB-SAM and U-Net in managing data scarcity. In our case, GB-SAM’s stable performance across varying dataset sizes offers an advantage in applications with limited annotated data, such as in the digital pathology field.

### 4.2. Assessing Model Generalizability across Diverse Datasets

After fine-tuning and evaluating the performance of the GB-SAM and U-Net models using 100% of the training data from the CRAIG dataset, we explored their ability to generalize to unseen data. The following section presents and discusses our findings, analyzing how effectively each model applies its learned segmentation capabilities to new images. Our analysis aims to highlight the strengths and weaknesses of GB-SAM and U-Net in terms of model generalization, offering insights into their practical utility and flexibility in digital pathology applications.

#### 4.2.1. Evaluating on GlaS

The GlaS test dataset consists of 37 benign and 43 malignant samples. In this section, we aim to evaluate the segmentation performance of the GB-SAM and U-Net models, trained on the CRAG dataset, across these categories.

Based on Table 4, we found that for benign areas, GB-SAM performs better in terms of the Dice coefficient and IoU, with average scores of 0.901 and 0.820, respectively. These metrics indicate that GB-SAM is highly effective in accurately identifying benign tumor areas, ensuring a strong match between the predicted segmentation and the ground truth. On the other hand, while still performing well, U-Net is slightly behind GB-SAM, with Dice and IoU scores of 0.878 and 0.797, respectively. However, it is noteworthy that U-Net outperforms GB-SAM in the mAP metric with an average score of 0.873, compared to GB-SAM’s 0.840. While U-Net may not match GB-SAM in segmentation precision and overlap, it has a small advantage in detecting relevant areas within benign contexts.

Now, when analyzing the performance on malignant tumors, the gap between GB-SAM and U-Net widens, particularly in the Dice and IoU metrics. GB-SAM maintains a lead with Dice and IoU scores of 0.871 and 0.781, respectively, versus U-Net’s 0.831 (Dice) and 0.745 (IoU). This indicates a consistent trend where GB-SAM outperforms U-Net in delineating tumor boundaries with greater precision, especially crucial in malignant tumors where accurate segmentation can significantly impact clinical outcomes. Interestingly, in the mAP metric for malignant tumors, U-Net (0.821) closes the gap with GB-SAM (0.796), suggesting that U-Net may be more adept at recognizing malignant features across a dataset, despite having lower overall segmentation accuracy. Visual segmentation results are shown in Figure 8 and Figure 9.

Both GB-SAM and U-Net exhibit strengths that make them valuable tools in the digital pathology domain. However, GB-SAM’s consistent accuracy and robustness across tumor types highlight its potential benefits for improved tumor segmentation and classification in clinical settings.

#### 4.2.2. Evaluating on Camelyon16

Lymph nodes present significant segmentation challenges, primarily due to often indistinct boundaries and the complexity of surrounding structures. In this context, analyzing the performance of GB-SAM and U-Net models, trained on the CRAG dataset, in segmenting lymph nodes within a subset of the Camelyon16 dataset offers valuable insights into their utility for complex pathological analyses.

Table 5 shows that GB-SAM outperforms U-Net across all metrics. Specifically, GB-SAM achieves a Dice score of 0.740, indicating a significantly higher degree of overlap between the predicted segmentation and the ground truth, in contrast to U-Net’s score of 0.491. This disparity suggests that SAM more effectively identifies lymph node boundaries within WSIs.

Similarly, GB-SAM’s IoU score of 0.612 exceeds U-Net’s 0.366, demonstrating that GB-SAM’s predictions more closely match the actual lymph node areas. Regarding mAP, GB-SAM leads with a score of 0.632 compared to U-Net’s 0.565. Although the gap in mAP between the two models is less pronounced than in Dice and IoU, GB-SAM’s higher score underlines its superior reliability in recognizing lymph nodes. Figure 10 shows a visual segmentation result.

Furthermore, even when trained on gland data from the CRAG dataset, GB-SAM’s excellent performance showcases its remarkable capacity for generalization to lymph node segmentation. This flexibility highlights GB-SAM’s robust and adaptable modeling approach, which can be used for different yet histologically related tissue types. In contrast, despite achieving high scores in segmenting gland structures, U-Net exhibits constraints in extending its applicability to other tissue types.

#### 4.2.3. Comparative Analysis: GB-SAM, SAM-Path, and Med-SAM

We summarize the results in Table 6. Despite a moderately lower IoU score of 0.813 compared to SAM-Path’s 0.883, GB-SAM obtained a higher Dice score of 0.900, which is essential for medical segmentation tasks. Notably, GB-SAM achieved a Dice score of 0.885 on the CRAG test dataset, surpassing SAM-Path’s 0.883, even though it used only 25% of the CRAG training data (see Table 2). This efficiency highlights GB-SAM’s capability to achieve high performance with limited training data, making it particularly suitable for scenarios with constrained data availability.

Compared to Med-SAM, which achieved a Dice score of 0.956 with a large training dataset, GB-SAM’s Dice score of 0.885 demonstrates a nominal difference. Despite the considerably smaller training dataset, this moderately high performance underscores GB-SAM’s effectiveness in generalizing from limited data to unseen cases. This ability to perform well with less data is critical for practical clinical deployment, where extensive annotated datasets may only sometimes be available.

Moreover, while we found no existing studies that use SAM for lymph node segmentation, GB-SAM’s performance in this task is noteworthy. Despite not being explicitly trained on lymph node data, GB-SAM’s acceptable segmentation performance indicates its potential to serve as a reliable tool across a diverse range of pathological tissues. This adaptability suggests that GB-SAM could be a valuable tool in clinical settings, offering robust segmentation capabilities across various medical images.

It is important to note that we did not run the experiments for SAM-Path and Med-SAM ourselves. Although their code is available on GitHub, replicating their experiments requires substantial computational resources. Therefore, we reported the statistics published in the original papers for comparison. This approach ensures that we provide a fair and accurate comparison based on the reported performance metrics of these models.

## 5. Conclusions

In this study, we adopted a new strategy of employing granular box prompts based on ground truth masks for fine-tuning our GB-SAM model, which is based on SAM. This approach aims to achieve more precise gland morphology segmentation, moving away from the traditional single-large box approach used in other works. This technique notably enhanced GB-SAM’s gland segmentation accuracy by supplying detailed data and mitigating ambiguity in regions with complex morphology.

Our experiments across the CRAG, GlaS, and Camelyon16 datasets showed that granular box prompts enable GB-SAM to focus on specific gland features, thus improving learning and generalization across various histopathological patterns. This method highlighted GB-SAM’s outstanding segmentation performance and adaptability, which is particularly helpful in digital pathology cases with limited data availability.

GB-SAM’s consistent performance and capability to generalize to new data, like lymph node segmentation, emphasize its potential for clinical applications. Although both GB-SAM and U-Net contribute valuable tools to digital pathology, GB-SAM’s robustness and success in extending beyond gland segmentation establish it as a strong option for tumor segmentation within the field of digital pathology.

Despite its promising performance, GB-SAM has some limitations. The Camelyon16 dataset is particularly challenging for segmentation due to the unclear boundaries and surrounding tissue structures in WSIs. As discussed in Section 4.2.2, the complexity of accurately detecting and segmenting lymph node metastases in Camelyon16 highlights the difficulty of handling images with complex details. Additionally, while we implemented early stopping based on validation loss to mitigate overfitting, maintaining state-of-the-art generalization capabilities across different datasets remains challenging. The variability in the complexity of tissue structures can impact GB-SAM’s performance, especially when segmenting highly complex or rare tissue structures.

To overcome these limitations, our future work will focus on enhancing the preprocessing pipeline to better standardize images and exploring advanced training strategies such as transfer learning and domain adaptation. We aim to ensure that GB-SAM can effectively generalize to a broader range of histopathological images, offering a promising path for its future development.

Addressing ethical and practical considerations is crucial for the responsible deployment of the GB-SAM model in clinical settings. This study represents the first phase of our project, focusing on testing the GB-SAM model’s segmentation capabilities. In the next phase, we will thoroughly address these considerations.

Future work will also include developing an interactive tool for pathologists and promoting the integration of GB-SAM into clinical workflows. This tool will enable medical professionals to interact with the model’s segmentation results, provide feedback, and validate its real-time performance.

While our study demonstrates GB-SAM’s robustness and generalization capabilities with reduced data, further evaluation on more diverse and extensive datasets is essential to capture its full performance for real-world applications. In future work, we plan to include additional datasets containing a more comprehensive range of histopathological variations and larger sample sizes to validate the generalizability of GB-SAM. This approach will help us confirm that our model is well suited for practical deployment in diverse clinical environments.

Moreover, while GB-SAM was explicitly trained for 2D segmentation tasks, we acknowledge the need to address the challenges associated with 3D medical imaging. Future work will focus on extending our approach to 3D data, exploring methods to process volumetric data efficiently, and ensuring consistency across slices.

GB-SAM shows great promise in digital pathology. Addressing its current limitations and expanding its validation to a wider range of datasets will be critical steps in its development. We look forward to enhancing GB-SAM’s capabilities and ensuring its robust performance in diverse clinical applications.

## Figures and Tables

**Figure 1 cancers-16-02391-f001:**
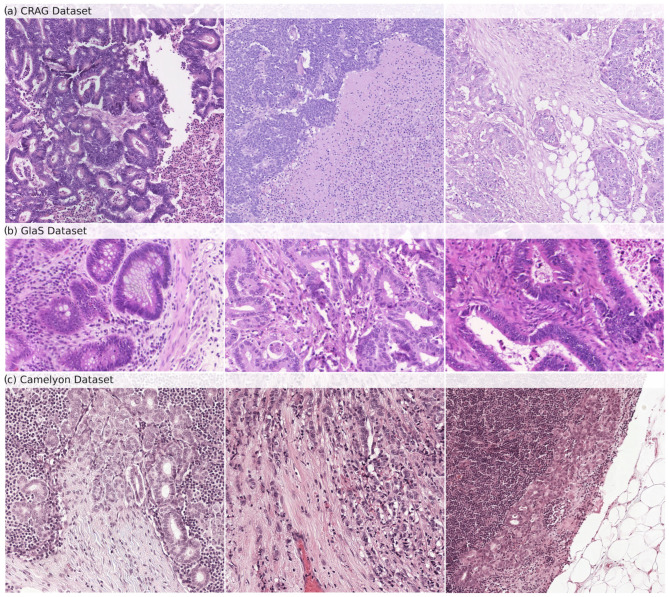
Samples of histopathological images from (**a**) CRAG, (**b**) GlaS, and (**c**) Camelyon datasets. These images show the diverse glandular structures and tissue types present in each dataset, which are used for training (CRAG) and evaluating (GlaS and Camelyon) GB-SAM.

**Figure 2 cancers-16-02391-f002:**
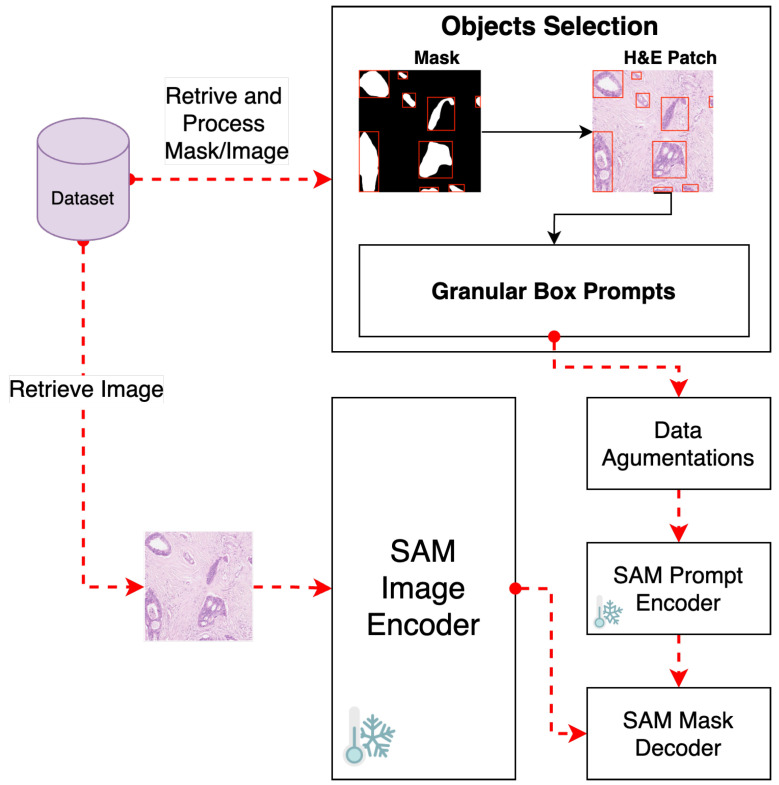
Pipeline for fine-tuning GB-SAM using granular box prompts.

**Figure 3 cancers-16-02391-f003:**
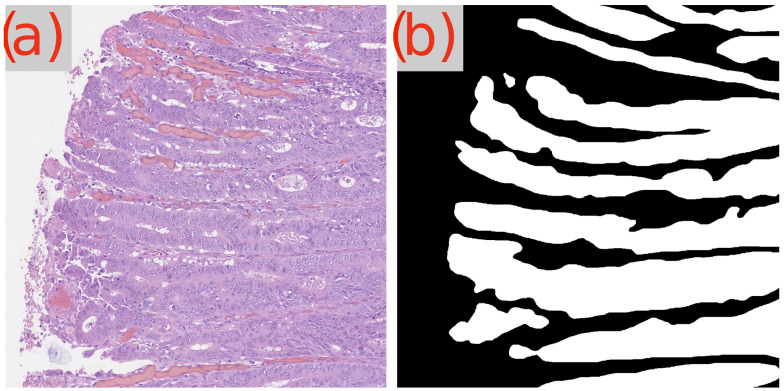
Segmentation results using GB-SAM and U-Net for image test_23 of the CRAG dataset: (**a**) H&E patch image, (**b**) ground truth mask, (**c**) U-Net predicted mask, (**d**) GB-SAM predicted mask.

**Figure 4 cancers-16-02391-f004:**
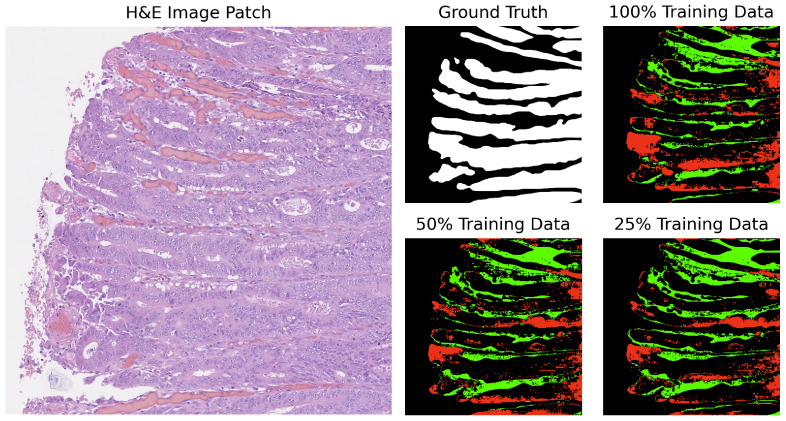
Segmentation results of GB-SAM on image test_23 of the CRAG dataset: red indicates underpredictions, and green indicates overpredictions relative to the ground truth mask.

**Figure 5 cancers-16-02391-f005:**
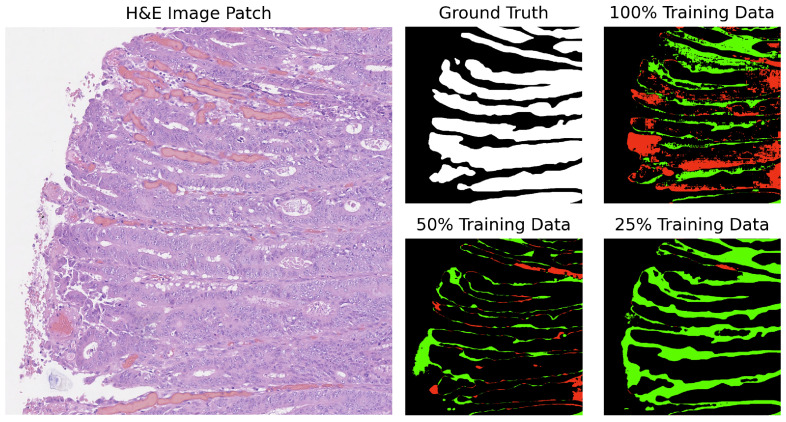
Segmentation results of U-Net on image test_23 of the CRAG dataset: red indicates underpredictions, and green indicates overpredictions relative to the ground truth mask.

**Figure 6 cancers-16-02391-f006:**
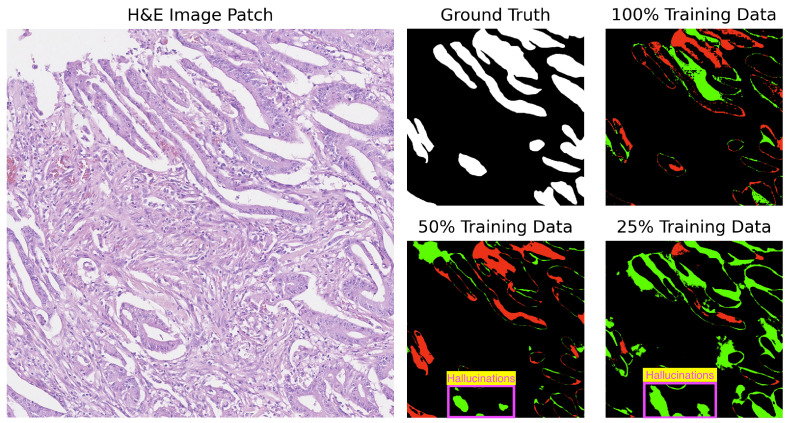
Segmentation results of U-Net on image test_15 of the CRAG dataset showing gland hallucinations. Red indicates underpredictions, and green indicates overpredictions relative to the ground truth mask.

**Figure 7 cancers-16-02391-f007:**
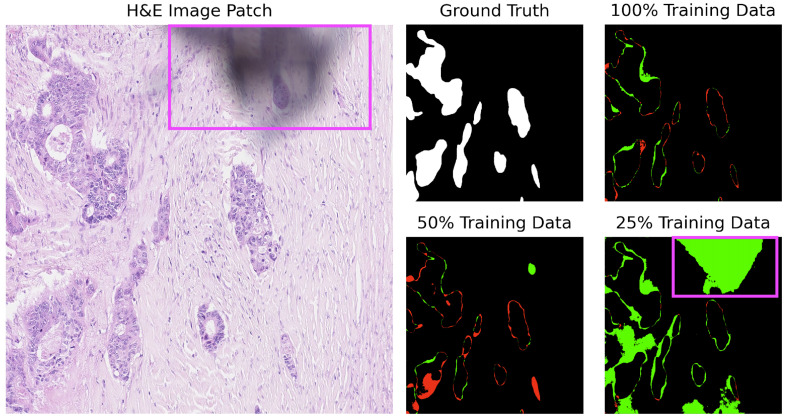
U-Net segmentation on image test_18 of the CRAG dataset: misclassification of digitization defects (purple square). Red indicates underpredictions, and green indicates overpredictions relative to the ground truth mask.

**Figure 8 cancers-16-02391-f008:**
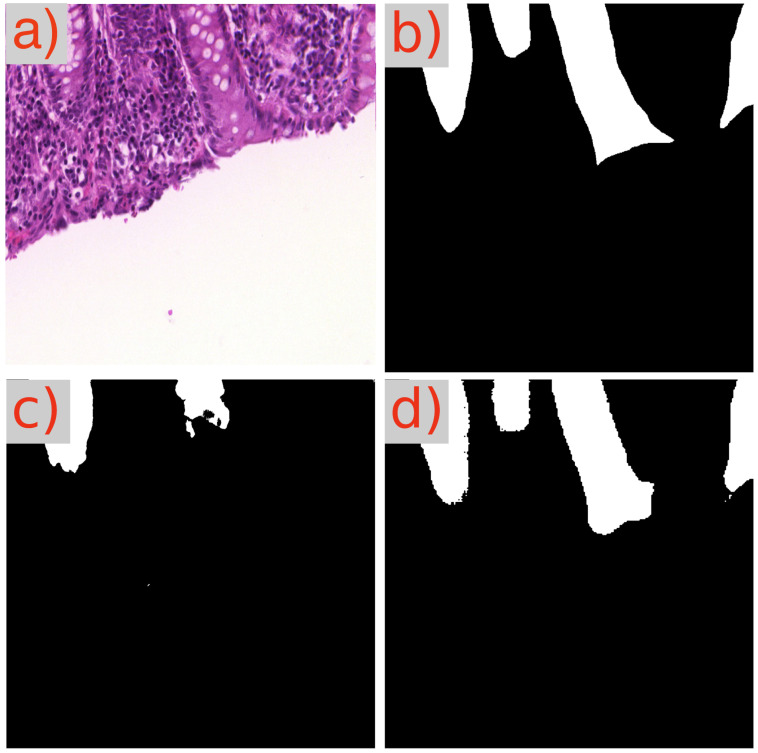
Segmentation results of GB-SAM and U-Net on a benign area in an image from the GlaS dataset: (**a**) H&E-stained patch image, (**b**) ground truth mask, (**c**) U-Net predicted mask, and (**d**) GB-SAM predicted mask.

**Figure 9 cancers-16-02391-f009:**
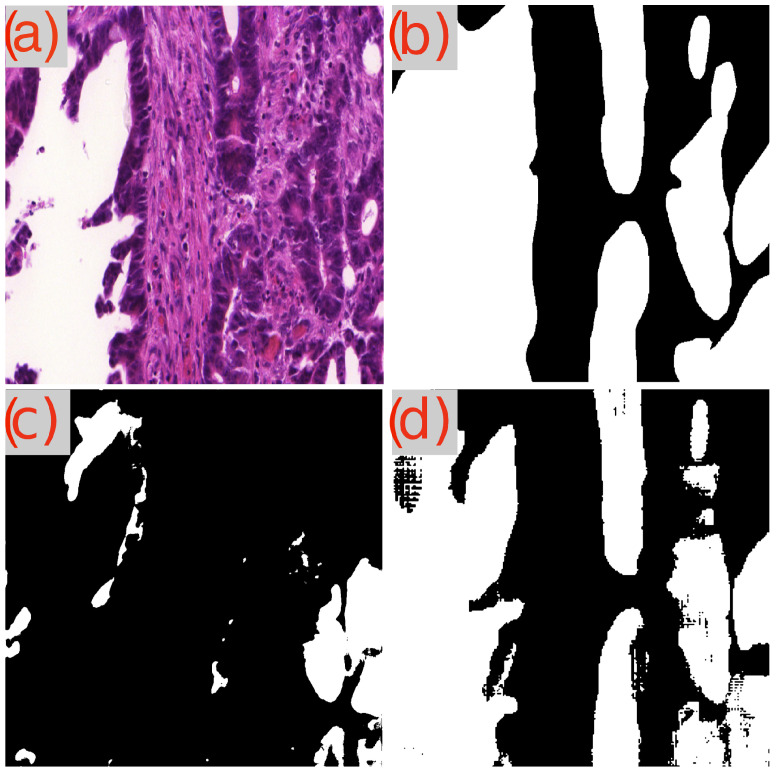
Segmentation results of GB-SAM and U-Net on a malignant area in an image from the GlaS dataset: (**a**) H&E-stained patch image, (**b**) ground truth mask, (**c**) predicted mask by U-Net, and (**d**) predicted mask by GB-SAM.

**Figure 10 cancers-16-02391-f010:**
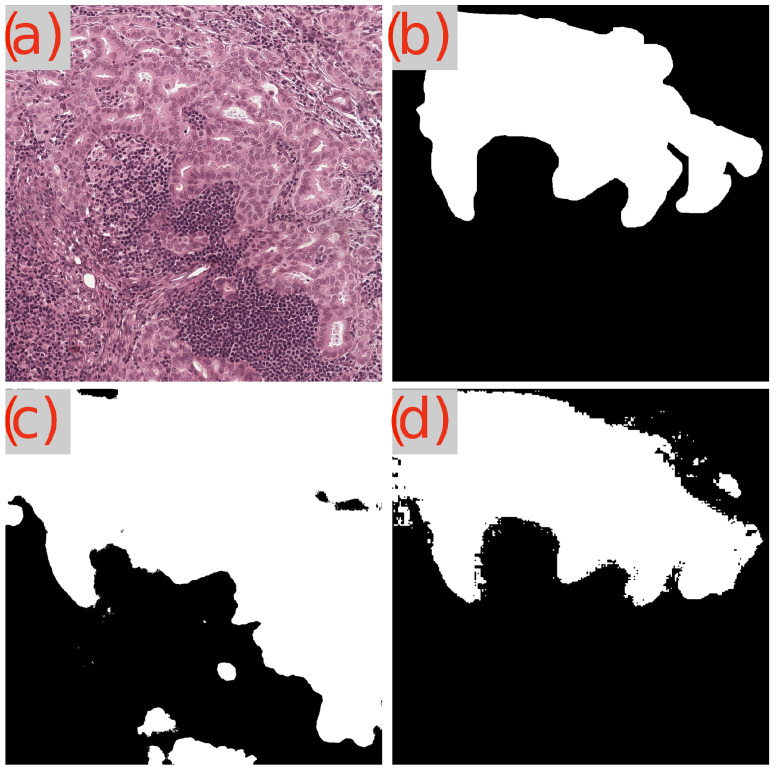
Segmentation Results of GB-SAM and U-Net on lymph node tumor in an image from the Camelyon dataset: (**a**) H&E-stained patch image, (**b**) ground truth mask, (**c**) predicted mask by U-Net, and (**d**) predicted mask by GB-SAM.

**Table 1 cancers-16-02391-t001:** Summary of studies on SAM models in medical image segmentation.

Study	Method	Dataset(s)	Key Findings	Limitations
Zhang et al. (2023) [16]	SAM-Path	BCSS, CRAG	Improvements in Dice score by 27.52% and IOU by 71.63% compared to vanilla SAM. Additional pathology foundation model further improves Dice by 5.07–5.12% and IOU by 4.50–8.48%.	Dependent on quality of trainable class prompts; complexity increases with additional models and fine-tuning.
Deng et al. (2023) [17]	SAM Zero-Shot	WSI	Good results on large objects; struggles with dense objects. Dice: Tumor—58.71–74.98, Tissue—49.72–96.49, Cell—1.95–88.30.	Ineffective for dense instance object segmentation; requires many prompts for better performance.
Liu et al. (2024) [18]	WSI-SAM	Histopathology images	Superior performance with multiresolution patches (Dice of 57.37); significant improvement in segmentation tasks.	Complexity in dual mask decoding; high computational resources required.
Ma et al. (2024) [19]	MedSAM	Multiple modalities	Outperforms specialist models; Dice: Various tasks—95.6% for colon gland segmentation, 96.5% for Skin cancer.	Requires large and diverse training datasets; high dependency on training data availability.
Ranem et al. (2024) [20]	SAM in radiology	Radiology, pathology	Improved segmentation accuracy; Dice for radiology: 84.49%; pathology: 39.05–77.80%.	Limited to specific medical imaging applications; need for robust annotation strategies.
Cui et al. (2024) [21]	All-in-SAM	Nuclei segmentation	Enhances SAM with weak prompts; competitive performance; Dice: 82.54%, IOU: 69.74%.	Dependent on quality of weak annotations; limited scalability for large datasets.

**Table 2 cancers-16-02391-t002:** Performance metrics of U-Net vs. GB-SAM at CRAG varying training dataset sizes, with superior results highlighted in green.

	Dice		IoU		mAP	
Training Size	GB-SAM	U-NET	GB-SAM	U-NET	GB-SAM	U-NET
100%	0.900	**0.937**	0.813	**0.883**	0.814	**0.904**
50%	0.876	**0.914**	0.781	**0.845**	0.778	**0.883**
25%	**0.885**	0.857	**0.793**	0.758	**0.788**	0.765
**SD**	**0.012**	0.041	**0.016**	0.064	**0.019**	0.075

**Table 3 cancers-16-02391-t003:** Performance minimal extremes for GB-SAM and U-Net models across different CRAG training dataset sizes.

Metric	Model	Train Size	Min. Score	Image
GB-SAM	**Dice**	100%	0.629	test_23
		50%	0.648	test_23
		25%	0.640	test_23
	IoU	100%	0.489	test_23
		50%	0.450	test_23
		25%	0.491	test_23
	mAP	100%	0.577	test_23
		50%	0.470	test_23
		25%	0.567	test_23
U-Net	Dice	100%	0.840	test_39
		50%	0.759	test_15
		25%	0.624	test_18
	IoU	100%	0.724	test_39
		50%	0.612	test_15
		25%	0.453	test_18
	mAP	100%	0.720	test_39
		50%	0.662	test_15
		25%	0.452	test_18

**Table 4 cancers-16-02391-t004:** Performance for GB-SAM and U-Net Models on GlaS dataset. Higher scores are highlighted in yellow.

Model	Grade	Metric	Average
GB-SAM	Benign	Dice	0.901
		IoU	0.820
		mAP	0.840
U-Net		Dice	0.878
		IoU	0.797
		mAP	0.873
GB-SAM	Malignant	Dice	0.871
		IoU	0.781
		mAP	0.796
U-Net		Dice	0.831
		IoU	0.745
		mAP	0.821

**Table 5 cancers-16-02391-t005:** Performance for GB-SAM and U-Net models on Camelyon16 dataset. Higher performance is highlighted in yellow.

Model	Metric	Average
GB-SAM	Dice	0.740
	IoU	0.612
	mAP	0.632
U-Net	Dice	0.491
	IoU	0.366
	mAP	0.565

**Table 6 cancers-16-02391-t006:** Performance comparison between GB-SAM, SAM-Path, and Med-SAM. Higher performance highlighted in yellow.

Model	Metric	CRAG	GlaS	Camelyon16
GB-SAM (Our model)	Dice	0.900	0.885	0.740
	IoU	0.813	0.799	0.612
	mAP	0.814	0.816	0.632
SAM-Path	Dice	0.884	-	-
	IoU	0.883	-	-
	mAP	-	-	-
Med-SAM	Dice	-	0.956	-
	IoU	-	-	-
	mAP	-	-	-

## Data Availability

All the datasets were downloaded from public domain. CRAG: https://warwick.ac.uk/fac/sci/dcs/research/tia/data/mildnet (accessed on 24 June 2024), Glas: https://datasets.activeloop.ai/docs/ml/datasets/glas-dataset/ (accessed on 24 June 2024), Camelyon: https://camelyon16.grand-challenge.org/Data/ (accessed on 24 June 2024).

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
