# Peer review of "Enhancing Medical Imaging Segmentation with GB-SAM: A Novel Approach to Tissue Segmentation Using Granular Box Prompts"

_cancers, 2024, doi:10.3390/cancers16132391_

Round 1

Reviewer 1 Report

Comments and Suggestions for Authors

The paper proposes the Granular Box Prompt Segment Anything Model (GB-SAM), an improved version of the Segment Anything Model (SAM) fine-tuned using granular box prompts with limited training data. The GB-SAM aims to reduce the dependency on expert pathologist annotators by enhancing the efficiency of the automated annotation process. Granular box prompts are small box regions derived from ground truth masks, conceived to replace the conventional approach of using a single large box covering the entire H&E-

stained image patch. This method allows a localized and detailed analysis of gland morphology, enhancing the segmentation accuracy of individual glands and reducing the ambiguity that larger boxes might introduce in morphological complex regions. The topic is interesting. The paper is well-written and structured.

The abstract: Could you please provide the specific numerical outcome from the results you obtained?

Please add that you compared also your proposed model to other versions of SAM.

Introduction:

Could you please add a paragraph describing the organization of the paper by the end of the introduction?

Related Works: Could you summarize the literature in a Table?

Also, add the related works’ numerical findings and limitations.

Materials and Methods

Could you please add more details to the dataset section?

Please add samples to the images included in the dataset

 The conclusion Kindly specify the constraints or drawbacks of your suggested model. Could you kindly provide information about your future plans or goals?

Reviewer 2 Report

Comments and Suggestions for Authors

The article, titled "Enhancing Medical Imaging Segmentation with GB-SAM: A Novel Approach to Tissue Segmentation Using Granular Box Prompts," presents a few shortcomings:

 1. The article emphasizes the performance of the GB-SAM model with reduced training data, highlighting its stability and generalization capabilities. However, the reliance on limited data might only partially capture the model's performance in more diverse and extensive datasets, which is crucial for real-world applications.

 2. While the article provides detailed performance metrics (Dice, IoU, mAP) for both GB-SAM and U-Net models, there are inconsistencies in the reported results. For instance, the performance of GB-SAM is superior in some metrics but not in others, which could indicate variability in the model's robustness across different datasets and conditions.

 3. The article mentions that U-Net tends to overfit when trained on smaller datasets, leading to increased over-predictions. This suggests that the GB-SAM model also faces similar issues, especially when applied to datasets with different characteristics than those used in the study.

 4. The study primarily focuses on the CRAG, GlaS, and Camelyon16 datasets. While these are well-known datasets, the need for external validation on other datasets limits the generalizability of the findings. The model's performance on a broader range of histopathological images still needs to be tested.

 5. The article must thoroughly discuss the GB-SAM model's limitations. For example, the potential challenges in segmenting highly complex or rare tissue structures still need to be addressed. A more comprehensive discussion of the model's limitations would provide a balanced view of its capabilities.

 6. The article does not address the ethical and practical considerations of deploying the GB-SAM model in clinical settings. Issues such as the need for expert validation, potential biases in the training data, and the implications of incorrect segmentation results on patient outcomes should be discussed.

 7. While the article compares GB-SAM with U-Net, it needs to provide a detailed comparison with other state-of-the-art segmentation models like SAM-Path and Med-SAM. A more extensive comparison would help understand GB-SAM's relative strengths and weaknesses.

 8. The figures and visual representations in the article, such as segmentation results and performance metrics, are crucial for understanding the model's performance. However, the clarity and consistency of these figures could be improved to convey the findings better.

 9. Given the authors' affiliation with the development of the GB-SAM model, there might be an inherent bias in reporting the results. Independent validation and replication of the study by other researchers would strengthen the credibility of the findings.

 10. SAM's pre-training on a vast dataset of natural images does not fully translate to medical images, which often have unique characteristics such as low contrast and ambiguous tissue boundaries—this lack of domain-specific knowledge results in subpar performance without additional fine-tuning for medical applications.

 11. SAM struggles with intricate medical segmentation tasks, such as distinguishing fine anatomical structures and complex tumor boundaries. This limitation is particularly evident in scenarios where even human interpretation is challenging, highlighting the need for more specialized models.

 12. SAM's performance is highly sensitive to the precision of the provided prompts. Minor deviations in bounding boxes or point prompts can significantly impair segmentation accuracy, making it less reliable for tasks requiring high precision.

 13. SAM's high computational cost and memory requirements, especially when applied to 3D volumetric data, limit its practical use in real-time medical applications. Models like FastSAM3D have been developed to address these issues, but they still need help in terms of efficiency and scalability.

 15. While SAM demonstrates strong zero-shot capabilities, its generalization to unseen medical data is sometimes unreliable. This is particularly problematic for medical applications where the model must adapt to various imaging modalities and pathological conditions.

 16. While promising, SAM's interactive segmentation capabilities are only sometimes effective for medical images. The model's reliance on user-provided prompts can lead to inconsistent results, especially in complex medical scenarios where precise segmentation is critical.

 17. SAM does not offer out-of-the-box fine-tuning capabilities, essential for adapting the model to specific medical tasks. This limitation necessitates additional development efforts to customize SAM for medical image segmentation.

 18. The first-stage segmentation in SAM can become a performance bottleneck, particularly in multi-stage pipelines. This issue can hinder the overall efficiency and effectiveness of the segmentation process in medical applications.

19. SAM's performance could be more effective for segmenting small objects, which is common in medical imaging. This limitation affects the model's ability to delineate small anatomical structures or pathological features accurately.

 20. Applying SAM to 3D medical imaging tasks involves processing each slice independently, leading to increased computational costs and potential inconsistencies across slices. This approach is less efficient compared to models specifically designed for volumetric data.

In summary, while SAM offers significant advancements in image segmentation, its application in medical imaging is limited by domain-specific knowledge gaps, sensitivity to prompt variations, computational constraints, and challenges in handling complex and small-scale medical tasks. These limitations highlight the need for further customization and fine-tuning to leverage SAM's potential in the medical field fully.

Comments on the Quality of English Language

minor

Round 2

Reviewer 2 Report

Comments and Suggestions for Authors

THANK YOU REVISION.

NO MORE COMMENTS